# The Metabolism of Glucosinolates by Gut Microbiota

**DOI:** 10.3390/nu13082750

**Published:** 2021-08-10

**Authors:** Kalina Sikorska-Zimny, Luciano Beneduce

**Affiliations:** 1Fruit and Vegetables Storage and Processing Department, Division of Fruit and Vegetable Storage and Postharvest Physiology, The National Institute of Horticultural Research, Pomologiczna 13a Street, 96-100 Skierniewice, Poland; 2Medical, Natural and Technical College, Institute of Health Sciences, Stefan Batory State University, Batorego 64c Street, 96-100 Skierniewice, Poland; 3Department of the Sciences of Agriculture, Food, Natural Resources, and Engineering (DAFNE) the University of Foggia, Via Napoli 25, 71122 Foggia, Italy; luciano.beneduce@unifg.it

**Keywords:** glucosinolates, diet, microbiota

## Abstract

Glucosinolates (GLS) and their derivatives are secondary plant metabolites abundant in *Brassicaceae*. Due to the enzymatic reaction between GLS and myrosinase enzyme, characteristic compounds with a pungent taste are formed, used by plants to defend themselves against insect herbivores. These GLS derivatives have an important impact on human health, including anti-inflammation and anti-cancer effects. However, GLS derivatives’ formation needs previous enzymatic reactions catalyzed by myrosinase enzyme. Many of the brassica-based foods are processed at a high temperature that inactivates enzymes, hindering its bioavailability. In the last decade, several studies showed that the human gut microbiome can provide myrosinase activity that potentially can raise the beneficial effects of consumption of vegetables rich in GLS. The variability of the human gut microbiome (HGM) in human populations and the diverse intake of GLS through the diet may lead to greater variability of the real dose of pro-healthy compounds absorbed by the human body. The exploitation of the genetic and biochemical potential of HGM and correct ecological studies of both isolated strains and mixed population are of great interest. This review focuses on the most recent advances in this field.

## 1. Introduction

Glucosinolates (GLS) are chemical compounds present in plants of the *Brassicaceae* family. This wide family includes many genuses, with some of them containing GLS (*Brassica* L., *Eruca* Mill. or *Sinapsis* L.). The richest in GLS among them is *Brassica*, with species like *Brassica oleracea* L. (cabbages, brussels sprouts, broccoli, cauliflower, collards, kale, kohlrabi), *Brassica rapa* L. (rape mustard, wild turnip, common mustard, Chinese cabbage, kale rape, pak choi), *Brassica carinata* A. Braun, *Brassica cretica* Lam., *Brassica elongata* Ehrh., *Brassica fruticulosa* Cirillo., *Brassica juncea* (L.) Czern., *Brassica napus* L., *Brassica nigra* (L.) W.D.J. Koch, *Brassica rupestris* Raf., *Brassica Ruvo* L.H. Bailey, and *Brassica tournefortii* [1,2].

These plants are widely popular and appreciated in cuisines all over the world. The edible part of these plants includes the leaves, flowers, roots, shoots, and seeds. *Brassicaceae* can be served as a dish in boiled, steamed, fried, stewed, or pickled form. These technological processes change not only the taste and form of the served plants but also the bioavailability of chemical compounds of brassicas to the human body [3], including GLS. GLS are well known to have beneficial effects for human health. They have anti-inflammatory and anti-cancer activity, act as an osteoarthritis-prevention factor, show neuroprotective and anti-obesity effects, and can reduce diabetes risk [4,5,6,7,8].

However, these properties are delivered by GLS derivatives, a wide group of pungent taste compounds that are produced by the plant against herbivores through the conversion of GLS (Figure 1). The enzymatic reaction between the myrosinase enzymes (EC 3.2.1.147, thioglucoside glucohydrolase, sinigrinase, sinigrase, MYR) and GLS leads to the production of GLS derivatives. The enzyme is present in plant tissue, stored in so-called myrosin cells/idioblasts—separately from the GLS, which are stored in the S cells. The enzymatic reaction occurs when the tissue is damaged by animals or insects [1,9]. The enzymatic hydrolysis takes place also in the human upper digestive tract during consumption (chewing) of raw plants. It needs to be remembered that MYR is sensitive to high temperatures and pH; consequently, if the plant was processed before ingestion (e.g., boiling in acidic condition), myrosinase is inactivated [10]. Therefore, these processes cause only partial absorption of GLS derivatives. GLS derivatives can be absorbed in the small intestine and the colon but only with the presence of MYR from plant tissue or human microbiota origin [11,12].

A large number of studies stated that the human gut microbiota can metabolise GLS with the production of bioactive compounds [9,13,14]. Several studies showed evidence of myrosinase-like activity and glucoraphanine hydrolysis—both in vitro and in vivo—to bioactive sulforaphane (SFN) by specific microbial strains, such as caecal microbiota. However, the knowledge of specific bacterial myrosinases is still limited to a few studies.

The first bacterial myrosinase was isolated from *Enterococcus cloacae* strain 506 [15] and had 71.8 kDa molecular weight. A second bacterial myrosinase (66 kDa m. w.) was purified from *Citrobacter* spp. strain Wye1 [16]. In that case, the enzyme was characterised as glycoside hydrolase family 3 (GH3) β-O-glucosidases and confirmed to be capable of producing isothiocyanates (ITCs) [15,16].

Additionally, two genes encoded for 6-phospho-β-glucosidases were identified on a pathogenic *E. coli* 0157:H7 strain able to metabolise GLS, but in that case, the enzyme’s activity was not fully characterised. In other studies, only purified protein extracts of different strains showed myrosinase activity even if a specific enzyme was not purified [17,18]. Finally, a myrosinase from a non-gut *Bacillus thuringensis* strain was partially purified and characterised by El Shora et al. (2016) [19].

Despite the high number of reports of myrosinase-like bacterial activities, there is a relatively low number of MYR characterisation studies and an observed variability of GLS metabolites produced. Some authors suggested that the lack of specific taxonomic correlations with ITC production previously reported may have been due to high interindividual variability in the taxa contributing to this activity [20]. Other authors suggest that the bacterial myrosinases mostly belonging to GH family 3 do not support complete hydrolysis or that isothiocyanates are not the major product of hydrolysis by microbiota [9]. It must be recalled that plant and aphids myrosinase are characterised as glucoside hydrolases GH family 1. Notwithstanding the limited knowledge of a precise enzymatic action of myrosinases of myrosinases-like bacterial enzymes, the conversion of GLS is confirmed in many studies. In several of them, when sinigrin was supplied, allyl-isothiocyanate (AITC) were among the detected metabolites [21,22]. In addition, glucoraphanin and glucoerucin were transformed to sulforaphane nitrile, erucin nitrile, and other metabolites [23].

## 2. Glucosinolates and Their Derivatives

Glucosinolates (GLS; thioglucosides) are water-soluble N-hydroxy sulphates that possess a sulphur-bound β-d-glucopyranose/β-thioglucose moiety and a sulfonated oxime. Differences in chemical formula among GLS are determined by different side chain derived from one of the amino acids [11,24,25].

The main division, according to the side chain, includes three groups of GLS:Aliphatic group from Met, Ala, Leu, Ile, and Val;Aromatic group from Phe and Tyr; andIndolic group from Trp.

The more common division is based on the chemical structure of aglycone [26]:
aliphatic GLS with methyl/2-propenyl GL—glucocapparin/sinigrin (Figure 2);thio-functionalised GLS with 4-methylsulfanylbutyl/4-methylsulfinylbutyl GL—glucoerucin/glucoraphanin (Figure 3); andindole-type GLS with 3-indolylmethyl GLS—glucobrassicin, gluconasturtiin, glucomoringin (Figure 4).

The biosynthetic pathways of GLS consist of different steps: the chain-elongation of amino acid precursors and the aliphatic/aromatic groups and the conversion of the oxime into the GLS structure, the chain-elongation stage (in the synthesis of the indolic group GLS this step is omitted), and alkylation, elimination, esterification, or oxidation of the aliphatic/indolic groups, dependent on the final compound [11,24].

GLS are potential precursors of valuable compounds that have pro-healthy properties, and their conversion is determined by hydrolysis through myrosinase. Several factors may influence the final product of GLS hydrolysis with myrosinase. The most important ones are the parent GLS or instability of thiohydroxamate-O-sulfonate aglycone, the pH of the environment, presence of ferrous ions or the epithiospecifier protein (ESP), nitrile-specific protein (NSP), or thiocyanate forming protein (TFP). As pointed out in Figure 1, GLS enzymatic hydrolysis can form appropriate derivatives dependently of the presence of certain cofactors and specific environmental conditions. The main products that can be generated are epithionitriles, isothiocyanates, nitriles, and thiocyanates [8,9,11,28] (Table 1).

GLS derivatives can be classified as follows:

Epithionitriles

These compounds are a result of enzymatic hydrolysis of GLS (if R contains double bond) but in the presence of ESP (epithiospecifier proteins) as a cofactor of myrosinase and pH < 6.5 [1,28,45]. The ESP presence not only promotes the formation of epithionitriles (and nitriles) but also a reduction of ITCs [46,47]. The mechanism is based on a reaction between proteins and the labile thiohydroximate-O-sulphate GLS aglucon. Finally, epithionitriles develops alkenyl GLS aglucons without spontaneously degradation to isothiocyanates [48,49]. The activity of ESP strongly depends on the presence of Fe^2+^ [1]. The possible mechanism uses Fe^2+^ to bind ESP by the amino acids: the subsequent insertion of the sulphur (followed by intramolecular transfer) into the terminal double bond to form the thiirane ring [49,50].

b.Nitriles

Nitriles formation occurs also in a presence of Fe^2+^ but at a lower pH (in a range 2–5) [1,45]. Some authors report the presence of a specific protein that can transform unstable GLS intermediate. This compound, called nitrile-specifier protein (NSP), is present only in specialist insect herbivores and diverts GLS hydrolysis to less toxic (for insects) nitriles instead of ITCs [13,49,51].

c.Isothiocyanates

These compounds are responsible for pungent taste (volatility and hydrophobicity of ITCs) [47,52]. This group includes compounds of the highest value for the human organism, like the more stable sulforaphane or less stable indole-3-carbinol [53,54]. Hydrolysis of ITCs from GLS is favoured in the absence of ESP [46]. -OH-isothiocyanates are formed in the enzymatic reaction of GLS if R contains β-hydroxylated sidechains and in neutral pH [1,47].

d.Thiocyanates

The thiocyanates are produced in the presence of the thiocyanate-forming protein (TFP), which is another ESP protein but exclusively from benzyl-; allyl-4 methylthiobutyl-GLS and in pH > 8 [1,47].

The beneficial effect of GLS breakdown into bioactive isothiocyanates is limited by the fact that most of the food-processing methods are largely inactivating plant enzymes. Therefore, absorption of almost intact GLS (with partial hydrolysis in the oral cavity during chewing process, in presence of plant MYR) in the human body may start in the stomach. The absorption of proper GLS breakdown product (with the presence of plant MYR) takes place in the small intestine. Since plant MYR activity, if any, is only residual in the upper digestive tract, the rest of GLS will be hydrolysed in the large intestine by the microbial MYR [11,55,56]. It needs to be remembered that many factors (plant variety, cultivation regimes, storage time and conditions, processing techniques, human microbiome) have a dramatic influence on bioactive compounds content as well as digestion processes and therefore bioavailability of GLS breakdown products [57].

## 3. Plant Composition and Human Gut Microbiome

The human gut microbiome composition and activity has important influence on the human metabolism, nutrition, physiology, and immune function. Therefore, the balance or imbalance microbiota is directly linked with the health and disease state of the host [58,59,60]. Among the most important functions of the human gut microbiome (HGM), the metabolization (fermentation) of indigestible component of diet is influenced by the quantitative and qualitative composition of the microorganisms that inhabit the gut. The HGM also plays a role in vitamin biosynthesis; metabolism of hormone-like bile acids, such as the regulation of the immune system (short-chain fatty acid (SCFA)—T-cell differentiation); protection from pathogens; and stress response [61,62,63]. Mota De Carvalho et al. (2018) [64] defined the gut microbiota as “the microbial population living in the gut, especially in the colon.”

This microbial population is characterized by a wide diversity both in terms of richness of species and their abundance (10^14^ cells, more than doubling the number of human cells) and represented by bacteria, archaea, and eukaryotes that live in an intimate relationship with the host. Despite this high diversity, approximately 93–98% of HGM is restricted to few phyla: Firmicutes, Bacteroidetes, Proteobacteria, and Actinobacteria. The most common genera found are *Bifidobacterium*, *Lactobacillus*, *Bacteroides*, *Clostridium*, *Escherichia*, *Streptococcus*, and *Ruminococcus*. It is worth mentioning that the features of HGM are different among individuals, caused by external (diet, health state, environment) and intrinsic factors (gender, age, genetic factors) [61,63].

A consequence of following a different diet with different sources of basic compounds from vegetable and/or animal derived food is the unique shaping of HGM for each individual. A correct diet and lifestyle can lead to eubiosis, when the gut ecosystem is well balanced, with higher microbial diversity. Improper diet and incorrect lifestyle generally affect the gut microbial diversity and can lead to dysbiosis, which is connected with a dramatic reduction of the beneficial microbial community and increase of Firmicutes/Bacteroidetes ratio [61,64]. This condition is associated with the occurrence of many diseases whose primary cause is inflammation, also regulated by the gut microbiota [61].

Scientific research confirms the effect of a vegetable-rich diet in maintaining good human health. Losasso et al. (2018) [65] found statistically significant higher richness in the vegetarian gut microbiota than in the omnivore one, including higher Bacteroidetes-related operational taxonomic units (OTUs). This influence is related to some bioactive plant-foods components and their role in shaping the balanced gut microbiota. To a general extent, the comparison between two basic feeding systems (vegetarian and non-vegetarian) characterized with high differences in the content of the meals (for vegetarian: high fibre and low calories; for non-vegetarian: high level of refined carbohydrates and saturated fatty acids) was conducted [61]. The results showed a final high-carbohydrate fermentation level for a vegetarian diet and a higher concentration of the products of amino acid fermentation for a non-vegetarian diet [66]. Different compounds provided in the vegetable-rich diet have a direct effect on the shaping of the gut microbiota. For example, fibre components are suitable substrates for Bacteroidetes metabolism that are able to produce short-chain fatty acids (SCFAs) [61,67,68]. These compounds are produced during bacterial fermentation of dietary fibre in the human gut [69,70]. Novel studies are pointing to the SCFAs’ role in inflammatory bowel diseases (IBD) and even in the immune-system response [71,72].

In opposition, a diet lacking vegetable sources stimulates the bacterial taxon incapable of digesting fibre (e.g., bile-tolerant: *Alistipes*, *Bilophila*, and *Bacteroides* spp.) and may induce the production of toxic catabolites by Firmicutes (*Roseburia* spp., *Eubacterium rectale*, and *Ruminococcus bromii*) [61,66]. This may lead to many pathologies connected with inflammation state; therefore, a vegetarian-diet-like form is more appropriate to maintain eubiosis in gut microbiota.

Some of the examples and one of the major compounds that influence HGM are the non-digestible carbohydrates (NDC), which include polysaccharides like cellulose, hemicellulose, and pectin [73]. NDC are defined as substrates suitable for bacterial fermentation in the colon, undigested in the upper gastrointestinal tract [74]. NDC classification is problematic due to a lack of straight division according to chemical formula, composition fermentation, or digestibility [75]. The ability of degradation of carbohydrates is characterised by approx. 10^3^ species of gut microorganism, although different bacteria degrade glycans in different ways. *Bacteroidetes* uses a series of proteins to bind, degrade, and import starch products. This degradation is conducted in the presence of glycan-degradation enzymes strategies also chosen by Firmicutes and Actinobacteria [67]. Fermentable dietary carbohydrates had an important role in *Faecalibacterium prausnitzii*, *Roseburia*, and *E. rectale* in mediating butyrogenic effect [68]. Certain types of carbohydrates contribute to the increase of the specific bacterial populations in the human gut, e.g., starch stimulates bifidobacteria, *Bacteroides* spp., *Ruminococcus bromii*, *E. rectale*, and *Roseburia* spp., β-glucan and fructan stimulate bifidobacteria (fructan also *Bacteroides*) [68].

Short-chain fatty acids (SCFAs) are one of the important fermentation products produced by anaerobic dietary fibre breakdown. Namely, acetate, butyrate, and propionate (in ratio 3:1:1 to 10:2:1) directly (and indirectly by decreasing the microenvironment pH) inhibit pathogens’ growth as well as the production of toxic compounds like amines or ammonia [64].

Prebiotics, according to the definition, are non-digestible (by the host) food ingredients that promote selective metabolism of beneficial microorganisms in the intestinal tract. The main probiotics (inulin, fructo-oligosaccharides—FOS, galacto-oligosaccharides, and lactulose) can be found in plant-source food. The healthy effect implies the ability to increase the number of *Bifidobacterium* but also lactobacilli and *Faecalibacterium prausnitzii* [68].

Another class of compounds with beneficial impact on human health are the polyphenols, which positively influence gut microbiota content by increasing the number of *Bifidobacterium* or lactobacilli but decreasing clostridial populations [68,76]. Certain polyphenol promotes different bacterial populations, e.g., ellagitannins affects Lachnospiraceae and Ruminococcaceae, while epicatechin/catechin significantly decreases *Clostridium perfringens* and *Clostridium difficile* [68,77].

## 4. Glucosinolates’ and Their Derivatives’ Influence on Human Gut Microbiota

Recent studies have been aimed at demonstrating that GLS can have an influence on the HGM and a significant, positive health impact on humans. In the case of functionally pro-active compounds like GLS, it is equally important to assess their influence in shaping the microbial community of the gut and to evaluate the active GLS metabolism of HGM towards the production of healthy compounds.

In a randomised crossover feeding study where the effects of a high-cruciferous vegetable diet on gut bacterial community profile was evaluated, it was shown that the gut bacterial composition differed significantly [78]. Moreover, *Eubacterium hallii*, *Phascolarctobacterium faecium*, *Alistipes putredinis*, and *Eggerthella* spp. were found as the microbial taxa closely associated with cruciferous vegetable intake. When broccoli were supplied with the diet in another controlled trial in studies that included both in murine and human testing, the HGM was affected, and changes in microbial populations were reported [79,80]. Particularly, in healthy adults, it was found that the consumption of broccoli determined a change in beta diversity primarily related to the positive change in the Bacteroidetes to Firmicutes ratio. At the same time, glucosinolate metabolites levels increased in plasma and urine, as reported by Charron et al. (2018) [81], supporting the hypothesis that changes in the microbiota induced changes in the availability of health-promoting glucosinolate metabolites. The encouraging findings of the above-reported feeding studies are limited by the diet in which cruciferous were implemented, where the effect of other foods cannot be separated from the GLS intake by the cruciferous vegetables. Therefore, a possible direct effect of GLS in shaping the gut microbiota is still under debate: in order to evaluate if an immediate effect of these changes was due only to GLS, Wu et al. (2019) [82] evaluated the correlation between the gut bacterial community composition and microbiota myrosinase-like activity. However, they found that one type of glucosinolate, sinigrin, had no effect on these activities, indicating that components other than glucosinolate could have an effect on microbial communities and their myrosinase-like activities. In a more recent study, with the selection of human faecal microbiome through in-vitro cultivation in the presence of brassica leachate, there was observed enrichment of lactic acid bacteria and *Enterobacteriaceae,* and particularly *Escherichia coli* strains isolated from this microbiome were capable of degrading GLS and also S-methylcysteine sulphoxide (another pro-healthy compound found in brassica) [83].

## 5. The Role of Microbial Communities in the GLS Metabolism in the Human Gut

In order to understand the fundamental role of microorganisms in human adsorption of prohealthy GLS derivatives, it is important to couple studies of HGM communities in vivo with studies ex vivo. Such a work should consider a specific strain that possesses MYR activity and constitutes a fundamental model for understanding the biochemistry of the process. Since the early findings of bacteria with myrosinase activity, a large number of isolated strains belonging to many taxonomically distinct clades have been isolated and characterised in vitro (Table 2). This shared ability is not surprising since glucosidase activity of Procaryotes is important both in the external environment (typically soil) and in digestive tracts of animals and insects due to its contribution to the turnover of organic matter and digestion of vegetal biomass.

In a pilot study, the association between glucosinolate metabolism and gut bacterial community composition, both in vivo and ex vivo, was examined [84] through an evaluation of urinary ITC excretion by high- and low-ITC excretion subjects. Interestingly, it was shown that glucoraphanin degradation by faecal bacteria ex vivo differed significantly between the faecal inoculated bacterial culture samples of the high- and low-ITC excretion. In contrast, the overall bacterial community structure did not differ significantly between the subjects either in faecal samples or in ex-vivo faecal culture samples. These findings support the thesis that complex microbial interactions, functional redundancy of HGM, and differential metabolic activity of stable microbial communities may be the key factor of healthy compounds’ production by gut bacteria rather than colonisation by key species exclusively associated with a brassica-rich diet.

The conducted studies showed differences in biochemical degradation of GLS conducted by bacteria. When evaluating the metabolites produced by different gut microbial strains, Luang-In et al., 2014 [85], stated that all tested bacteria metabolised glucoerucin to erucin and erucin nitrile (NIT). Differences among strains were found in the degradation of glucoiberin and glucoraphanin (from 10–20% to 80–90%), producing erucin, erucin NIT, iberverin, and iberverin NIT from the two GSLs [85]. Another study in an ex-vivo experiment consisting of incubating glucoraphanin in contact with the anaerobic gut microbiota of rats demonstrated that cecum microbiota of rats can hydrolyse glucoraphanin in sulforaphane [86].

Moreover, by evaluating specific human gut-isolated strains, Luang-In et al. (2016) [85] proved that *Enterococcus casseliflavus* CP1 and *Escherichia coli* VL8 can metabolise glucoiberin and glucoraphanin, producing distinct ITC and nitriles. The authors also suggested that the metabolism of GLS in human gut bacteria is a more complex phenomena since different bacterial strains may have specific mechanisms in metabolising GLS. The same authors in a further study conducted with different GLSs and relative desulfo-GLS confirmed the production of distinct, strain-specific ITC and nitriles. Additionally, the production of nitriles from desulfo-GLS suggests an alternative metabolism via desulfation for the food-based GSLs [91].

According to Tian et al. (2018) [9], the hypothesis that differences in bacterial microflora between individuals may lead to inter-individual variation in the extent to which GLSs are hydrolysed has received increasing confirmation.

More recently, for the first time, the genetic and biochemical basis for activation of glucosinolates to isothiocyanates were described by Liou et al. (2020) [96]. In this study, a gut commensal strain of *Bacteroides thetaiotaomicron* was used, and an operon required for glucosinolate metabolism was identified. In vitro and in-vivo experiments on mice, also including a transformed non-metabolising relative and deleted mutant of the strain, confirmed ITC production when operon was present. Finally, by examining human stool samples and sequencing data from healthy individuals, the gene cluster was present in >40% of all individuals, suggesting that this operon is typical in humans.

Despite this important progress towards a deeper understanding of the mechanisms that lead to ITC production by gut bacteria, there is still a lack of knowledge about the alternative pathways that may lead to ITC bacterial catabolism and consequent reduction of the ITC adsorption by human intestinal cells (Figure 5) [13,97]. When considering the negative effect of ITC on bacteria [98] and recent findings of herbivorous insect gut bacteria able to degrade ITC [99] rapidly, it is not surprising that the role of the gut microbiome is ambivalent. As the consequence, the increase of ITC absorption by the human gut is a result of the balance of the different biochemical pathways.

The most recent findings of a GLS metabolism operon [96] together with a promising metabolic-engineering approach [100] could effectively contribute to the overcoming of unwanted ITC loss due to the side-metabolism of GLS and ITC by gut microbiota and increase the efficiency of absorption by human cells.

## 6. Conclusions

GLS are well known as pro-healthy compounds. Their abundance in *Brassicaceae* have been confirmed in many studies as well as their beneficial impact on the human health. The most active isothiocyanates are GLS derivatives—produced after hydrolysis by myrosinase enzymes. Unfortunately, most of the food-processing treatments inactivate the MYR enzyme, hampering the production of bioactive ITC. However, the conversion process of GLS to its derivatives may occur not only in the presence of MYR from plants but also from bacteria. Therefore, the human microbiome can have a significant impact on increasing the bioavailability of GLS derivatives.

Recent advances in the ecology of the human gut microbiome and the study of GLS metabolism in this complex environment revealed important mechanisms influencing the bioavailability of ITCs in food. Currently, it is very important in this area of research to determine (and evaluate) the factors that shape HGM and have a positive effect on the conversion of GLS to ITC. In addition, further studies should aim at assessing the factors that drive bacterial conversion of GLS to other compounds, such as nitriles, that do not have pro-healthy effects. According to other authors’ [45] metabolomics studies, a multi-omics approach will allow to evaluate the complex chemical pool of products from GLS derived by bacterial metabolism in the complex HG environment. This integration of methods is the most promising approach to address most of the open questions.

Additionally, future research in the field of metabolic engineering on the most promising isolated strain may be used to obtain new types of probiotics but also symbiotics to exploit the full potential of brassicas plants and their bioactive derived compounds.

## Figures and Tables

**Figure 1 nutrients-13-02750-f001:**
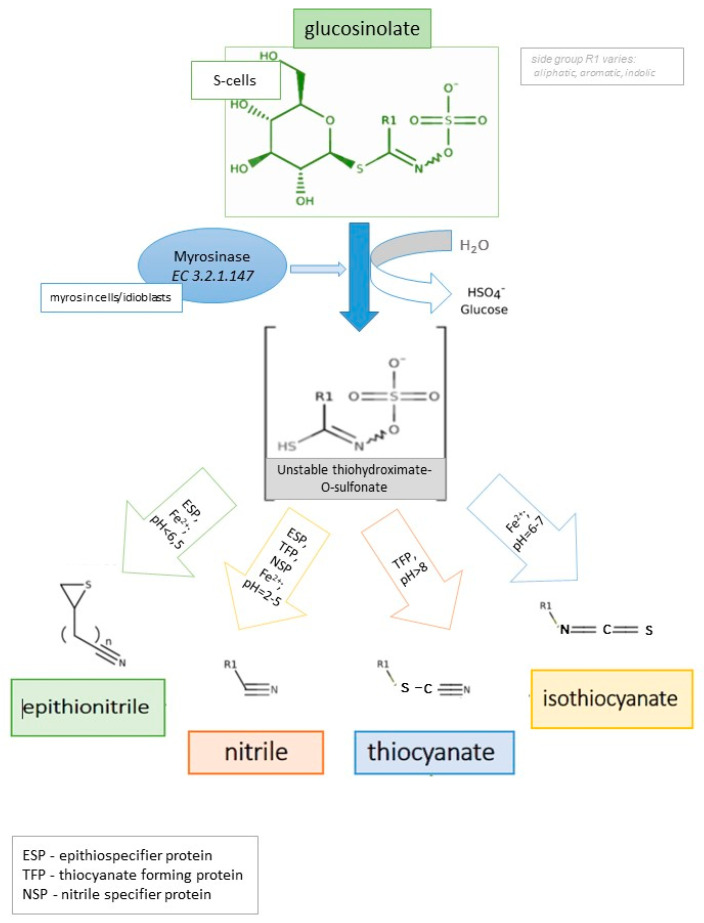
Scheme of enzymatic hydrolysis of GLS and their derivatives [1,11,13].

**Figure 2 nutrients-13-02750-f002:**
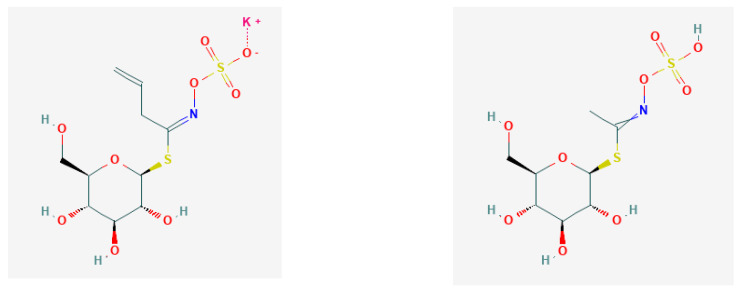
Chemical structures of glucocapparin (PubChem Identifier: *CID* 5281133; https://pubchem.ncbi.nlm.nih.gov/compound/21600408) and sinigrin (PubChem Identifier *CID* 23682211; https://pubchem.ncbi.nlm.nih.gov/compound/23682211), respectively, from PubChem [27].

**Figure 3 nutrients-13-02750-f003:**
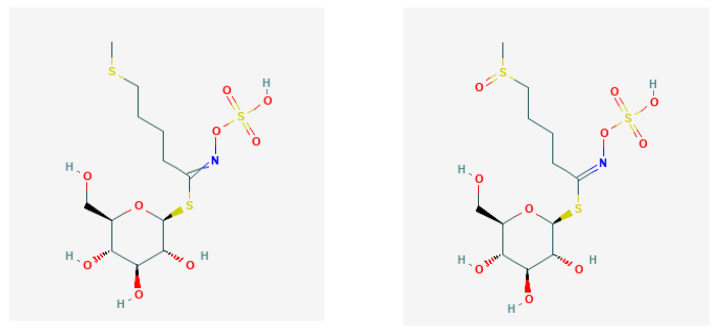
Chemical structures of glucoerucin (PubChem Identifier: *CID* 656539; https://pubchem.ncbi.nlm.nih.gov/compound/656539) and glucoraphanin (PubChem Identifier: *CID* 9548634; https://pubchem.ncbi.nlm.nih.gov/compound/9548634), respectively, from PubChem [27].

**Figure 4 nutrients-13-02750-f004:**
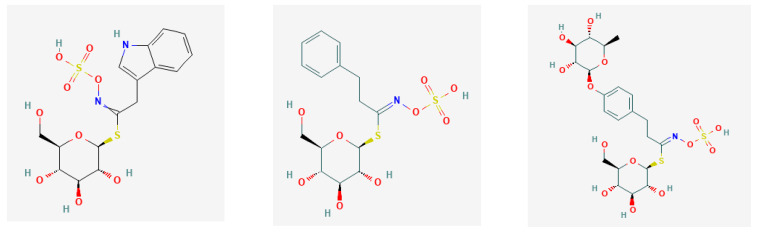
Chemical structures of glucobrassicin (PubChem Identifier: *CID* 656506; https://pubchem.ncbi.nlm.nih.gov/compound/656506), gluconasturtiin (PubChem Identifier: *CID* 5464032; https://pubchem.ncbi.nlm.nih.gov/compound/5464032), and glucomoringin (PubChem Identifier: *CID* 102222710; https://pubchem.ncbi.nlm.nih.gov/compound/102222710), respectively, from PubChem [27].

**Figure 5 nutrients-13-02750-f005:**
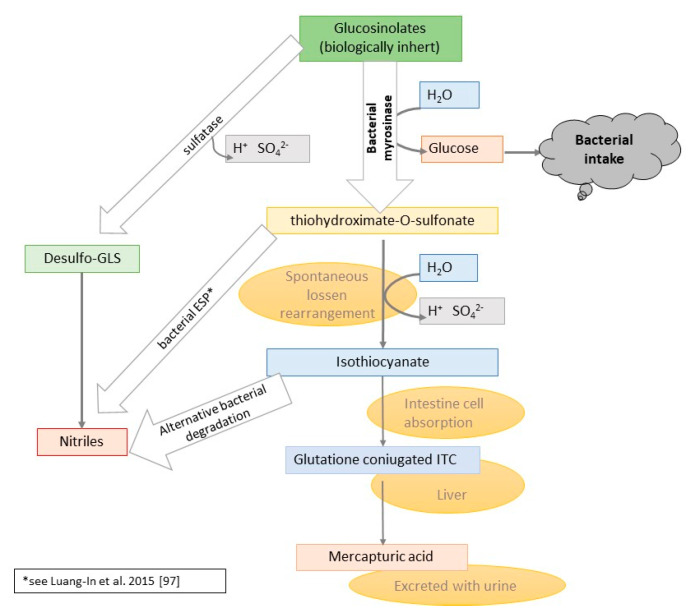
Scheme of GLS breakdown products by human microbiota and their adsorption in the large intestine. The steps directly involving bacterial activity are evidenced in bold.

**Table 1 nutrients-13-02750-t001:** Glucosinolates and examples of their possible derivatives.

Trivial Names/Abbreviation	Semisystemic Names *	Possible Derivatives *	Ref.
Aliphatic			
Epiprogoitrin/EPI	2(*S*)-2-Hydroxy-3-butenyl	(S)- and (R)-1-Cyano-2-hydroxy-3-butene	[29]
Glucoalyssin/GAL	5-Methyl-sulfinyl-pentyl	no corresponding isothiocyanate (ITC)/indole	[30]
	5-Methylsulfinylpentyl		[31]
Glucobrassicanapin/GBN	4-Pentenyl	5-hexenenitrile	[32]
	Pent-4-enyl		
Glucoberteroin/GOP	5-Methylthiopentyl	6-(Methylsulfanyl) hexanenitrile	[31]
Glucoerucin/GER	4-Methlythio-butyl	1-isothiocyanato-4-methylsulfanylbutane; erucin	[33]
Glucoerysolin	4-Methyl-sulfonyl-butyl	4-(Methylsulfonyl)pentane nitrile	[34]
	4-Methylsulphonylbutyl		
Glucoiberin/GIB	3-Methyl-sulfinyl-propyl	3-Methylsulfinylpropyl isothiocyanate; iberin;	[35]
Glucoibervirin/GIV	3-Methylthio-propyl	3-methylthiopropyl isothiocyanate	[36]
	3-Methylthiopropyl		
Gluconapin/GNA	3-Butenyl	3-butenyl isothiocyanate	[37]
	But-3-enyl		
Gluconapoleiferin/GNP	2(*R*)-2-Hydroxy-4-pentenyl	respective oxazolidinethiones	[38]
	2-Hydroxypent-4-enyl		
	2-Hydroxy-pent-4-enyl		
Glucoraphanin/GRA	4-Methyl-sulfinyl-butyl	1-Isothiocyanato-4-(methylsulfinyl)butane; sulphoraphane	[33]
	4-Methylsulfinylbutyl;		
Glucoraphenin/GRE	4-Methyl-sulfinyl-3-butenyl	1-Isothiocyanato-4-(methylsulfinyl)butane; sulphoraphane	[31]
	4-Methylsulfinylbut-3-enyl;		
Glucorapiferin/GRPProgoitrin/PRO	2(*R*)-2-Hydroxy-3-butenyl	(S)- and (R)-1-Cyano-2-hydroxy-3-butene; crambene	[29]
Sinigrin/SIN	2-Propenyl	3-isothiocyanatoprop-1-ene; allyl isothiocyanate	[39]
	Prop-2-enyl		
Indolic			
1-hydroxy-3-indolyl methyl			
4-Hydroxyglucobrassicin/4OHBGS	4-Hydroxy-3-indolyl-methyl;	4-hydroxy-3-indoleacetonitrile	[40]
	4-Hydroxyindol-3-ylmethyl;		
	4-Hydroxy-3-indolylmethyl		
4-Methoxyglucobrassicin/4MEGBS	4-Methoxy-3-indolyl-methyl;	4-methoxyindolyl-3-acetonitrile	[41]
	4-Methoxyindol-3-ylmethyl;		
	4-Methoxy-3-indolylmethyl		
Glucobrassicin/GBS	3-Indolyl-methyl;	indole-3-carbinol	[33]
	3-Indolylmethyl;		
	Indol-3-ylmethyl		
Neoglucobrassicin/NGBS	1-Methoxy-3-indolyl-methyl;	N-methoxy indole- 3-carbinol	[42]
	N-Methoxyindol-3-ylmethyl;		
	N-Methoxy-3-indolylmethyl		
Aromatic			
Glucobarbarin/GBA	2(*S*)-2-Hydroxy-2-phenyl–ethyl	p-hydroxyepiglucobarbarin	[43]
	(2S)-2-Hydroxy-2-phenethyl	(R)-barbarin; (R)-resedine; 3-Hydroxy-3-phenylpropanenitrile	[44]
Gluconasturtiin/GNR	2-Phenyl–ethyl	phenethyl isothiocyanate	[39]
	Phenethyl	phenyl-3-propanenitrile	[32]
	2-Phenethyl		
Glucotropaeolin/GTL	Benzyl	isothiocyanatomethylbenzene; benzyl ITC	[31]

* PubChem-chemical formulas [27].

**Table 2 nutrients-13-02750-t002:** Diversity of bacterial strains capable of in-vitro metabolism of GLS (modified from Narbad and Rositer 2018) [13]. Substrate abbreviations are reported according to the abbreviation used in Table 1.

Phylum	Family	Genus	Species (Strain)	Substrate	Products		Cell-Free Protein Extract *	Reference
				GSL	ITC	NIT		
Actinobacteria	Bifidobacteriaceae	*Bifidobacterium*	*pseudocatenulatum*	SIN, GTL	NT ^‡^	NT		[87]
			*adolescents*	SIN	−	+	1	
			*adolescents*	GTL	−	+		
			*longum*	SIN, GTL	−	NT		
Bacteroidetes	Bacteroidaceae	*Bacteroides*	*thetaiotaonicron (II8)*	SIN	+	−		[22]
Firmicutes	Bacillaceae	*Bacillus*	*cereus*	rape seed meal	+	NT		[88]
			*sSubtilis*	PRO	+	NT		[89]
		*Bacillus (isolates)*	*spp.*	SIN	NT	NT		[90]
	Enterococcaceae	*Enterococcus*	*casseliflavus CP1*	SIN	+	+	2	[91]
				GER	+	+		[85]
				GIB	Trace	−		
				GRA	−	Trace		
				GTL	+	+		[91]
				GNR	+	+		[91]
	Lactobacillaceae	*Lactobacillus*	*spp.*	SIN	NT	NT		[90]
			*plantarum KW30*	GRA, GIB	−	+		[29]
			*gasseri*	GRA	−	+		[86]
			*acidophilus*	GRA	−	+		
			*casei*	GRA	−	+		
			*plantarum*	GRA	−	+		
			*curvatus (various strains)*	SIN	NT	NT		[92]
			*plantarum (various strains)*	SIN	NT	NT		
			*(LEM)*	SIN	NT	NT		[93]
			*(LEM)*	PRO	NT	NT		
			*agilis R16*	SIN	+	+	2	[21,91]
				GER	+	+		[85]
				GIB	−	−		
				GRA	−	−		
				GTL	+	+		[91]
				GNR	+	−		
	Streptococcaceae	*Lactococcus*	*lactis subsp.lactis KF147*	GRA, GIB	−	+		[29]
	Listeriaceae	*Listeria*	*monocytogenes*	SIN	+	NT		[94]
			*monocytogenes*	SIN	+	NT		[95]
	Lactobacillaceae	*Pediococcus*	*pentosaceus*	SIN	NT	NT		[92]
			*acidilactici*	SIN	NT	NT		
			*pentosaceus*	SIN	+	NT		[94]
	Staphylococcaceae	*Staphylococcus*	*carnosus (various strains)*	SIN	NT	NT		[92]
			*spp.*	SIN	NT	NT		[90]
			*epidermis*	PRO	+	NT		[89]
			*aureus*	SIN	+	NT		[94]
			*carnosus*	SIN	+	NT		
		*Streptomyces*	*(isolates)*	SIN	NT	NT		[90]
Proteobacteria	Enterobacteriaceae	*Aerobacter (Klebsiella)*	*aerogenes*	PRO	+	NT		[89]
		*Citrobacter*	*WYE1*	SIN	−	−	3	[16]
		*Enterobacter*	*cloacae*	SIN	NT	NT	4	[15]
			*cloacae KS50*	SIN	NT	NT	4	[17]
		*Escherichia*	*coli VL8*	SIN	+	+	2	[91]
				GER	+	+		[85]
				GIB	+	+		
				GRA	+	+		
				GTL	+	+		[91]
				GNR	+	+		
		*Escherichia*	*coli Nissle 1917*	GRA, GIB	−	+		[29]
			*coli*	PRO	+	NT		[89]
			*coli O157:H7*	SIN	+	NT		[18]
				SIN	+	NT		[94]
			*fecalis*	SIN	+	NT		
		*Salmonella*	*typhimurium*	SIN	+	NT		
			*spp.*	SIN	+	NT		[95]
		*Paracolobactrum **	*aerogenoides*	PRO	+	NT	5	[89]
	Morganellaceae	*Proteus*	*vulgaris*	PRO	+	NT		
	Pseudomonadaceae	*Pseudomonas*	*spp.*	SIN	NT	NT		[90]
			*fluorescens*	SIN	+	NT		[94]

1 = ITC produced by cell-free extract. 2 = SIN degraded but no products found in cell-free extract. 3 = SIN, GTL, GRA, GER degraded by cell-free extract; MYR activity confirmed in vitro. 4 = SIN degraded by cell-free extract; MYR activity confirmed in vitro. 5 = PRO degraded by cell-free extract; ITC found. ^‡^ NT, not tested. * invalid name in the standing bacterial nomenclature, not reassigned to a taxonomically valid genus to date.

## Data Availability

Not applicable.

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
