# Peer review of "The Metabolism of Glucosinolates by Gut Microbiota"

_nutrients, 2021, doi:10.3390/nu13082750_

Round 1
Reviewer 1 Report
General comments
The new version of the manuscript entitled "The metabolism of glucosinolates by gut microbiota" has been significantly improved and, some changes are still required.
The following points are the aspects that still need to be changed:
Abstract
Line 12- 13: Remove the words “, generally most” the correct sentence is “Glucosinolates (GLS) and their derivatives are secondary plant metabolites abundant in Brassicaceae.”
Lines 16-18: replace the sentence “Although this effect may be obtained in the presence of an enzyme (enzymatic reaction of GLS) and many of the brassica-based food is treated with a high temperature, that inactivates enzymes.” for “However, GLS derivatives formation need previous enzymatic reaction canalized by mirosynase enzyme, but many of the brassica-based food are treated at high temperature that inactivates enzymes, hindering its bioavailability.” to better understanding.
Introduction
Line 30: Remove the word “mostly” the correct sentence is “Glucosinolates (GLS) are chemical compounds present in plants of the Brassicaceae family.”
Line 44: change the word “coupled” for “attributed”
Line 51: add the word “chewing” in the sentence “…during consumption (chewing) of raw plants.” to be clearer.
Line 54: Finish the sentence "These processes cause only partial absorption" e.g. "These processes cause only partial absorption of GLS derivatives"
Line 81: suppress the comma between detected and metabolites
Line 137: “c. thiocyanates“ should be “c. Isothiocyanates“
Lines 161: Change the sentence “The process of food digestion is influenced by the quantitative and qualitative composition of the microorganisms that inhabit the gut.” for “The metabolization (fermentation) of indigestible component of diet is influenced by the quantitative and qualitative composition of the microorganisms that inhabit the gut.” It seems that the authors heve still not clear about the concept of digestion!
Line 164-166: The sentence “Among these functions, the so-called human gut microbiome (HGM) is fundamental for the further fermentation of food components (digestion and energy recovery), and has a key role in vitamin biosynthesis...” must be changed for other like this “In addition to the fermentation of the indigestible component of the diet, the so-called human gut microbiome (HGM) has a key role in vitamin biosynthesis…” Please learn about the difference within digestion of food and fermentation of food compounds by microbiota!!!!
- Finally, in the reviewer’s opinion, for better tracking and understanding of the manuscript, Figure 2 should be before than Table 1.
Author Response
We are pleased to submit for evaluation our review titled “The metabolism of glucosinolates by gut microbiota.” (by Kalina Sikorska-Zimny and Luciano Beneduce), which has gone under the review in Nutrients, has been rejected from publication (December 2020), than editors encourage authors for publication (May 2021).
Note: "This manuscript is a resubmission of "nutrients-1027223" for special issue "Gut Microbiota in Human Health and Diseases".
Below we had answered all remarks made by the Reviewers.
In the body of manuscript according to editor’s demanding we have left all previous marked changes and made a revisions to the manuscript, marked up using the “Track Changes” function.
K.Sikorska-Zimny
L.Beneduce
General comments
The new version of the manuscript entitled "The metabolism of glucosinolates by gut microbiota" has been significantly improved and, some changes are still required.
Thank you for your comment. We have changed many parts of the manuscript according to all Reviewers comments.
The following points are the aspects that still need to be changed:
Abstract
Line 12- 13: Remove the words “, generally most” the correct sentence is “Glucosinolates (GLS) and their derivatives are secondary plant metabolites abundant in Brassicaceae.”
Lines 16-18: replace the sentence “Although this effect may be obtained in the presence of an enzyme (enzymatic reaction of GLS) and many of the brassica-based food is treated with a high temperature, that inactivates enzymes.” for “However, GLS derivatives formation need previous enzymatic reaction canalized by mirosynase enzyme, but many of the brassica-based food are treated at high temperature that inactivates enzymes, hindering its bioavailability.” to better understanding.
Introduction
Line 30: Remove the word “mostly” the correct sentence is “Glucosinolates (GLS) are chemical compounds present in plants of the Brassicaceae family.”
Line 44: change the word “coupled” for “attributed”
Line 51: add the word “chewing” in the sentence “…during consumption (chewing) of raw plants.” to be clearer.
Line 54: Finish the sentence "These processes cause only partial absorption" e.g. "These processes cause only partial absorption of GLS derivatives"
Line 81: suppress the comma between detected and metabolites
Line 137: “c. thiocyanates“ should be “c. Isothiocyanates“
Lines 161: Change the sentence “The process of food digestion is influenced by the quantitative and qualitative composition of the microorganisms that inhabit the gut.” for “The metabolization (fermentation) of indigestible component of diet is influenced by the quantitative and qualitative composition of the microorganisms that inhabit the gut.” It seems that the authors heve still not clear about the concept of digestion!
Thank you for pointing misleading parts, we have corrected appropriate words and sentences.
We organized the first part of the paragraph to better explain the roles of HGM and interaction with the host. We omitted the term “digestion” that was somewhat misleading the interpretation of the sentence.
Line 164-166: The sentence “Among these functions, the so-called human gut microbiome (HGM) is fundamental for the further fermentation of food components (digestion and energy recovery), and has a key role in vitamin biosynthesis...” must be changed for other like this “In addition to the fermentation of the indigestible component of the diet, the so-called human gut microbiome (HGM) has a key role in vitamin biosynthesis…” Please learn about the difference within digestion of food and fermentation of food compounds by microbiota!!!!
According also to the previous comment, we have entirely changed the order of the paragraph and corrected the misleading parts. The concepts are now clearly reported, as the Reviewer can also see from the scientific literature that has been studied and, consequently, cited in the manuscript.
From reference 60: “By definition, the dietary fiber goes through the small intestine, reaches the colon, and can be used by fiber-degrading members of the microbiota. The main results of such a metabolism are short-chain fatty acids (SCFAs), namely, acetate, propionate, and butyrate, which have recognized health-promoting activities, such as anti-inflammatory, anticarcinogenic, and immune-regulatory functions”.
From reference 61: “The eubiotic state is characterized by high microbial diversity, with a marked prevalence of microbes potentially beneficial to us, as they influence practically all functions in the intestine: digestion and energy harvest, mucosal immunity, integrity of the intestinal barrier, protection from pathogens, production of vitamins and other useful metabolites, such as the short chain fatty acids (SCFAs).
From reference 63: “…These include the fermentation of indigestible food components into absorbable metabolites, the synthesis of essential vitamins, the removal of toxic compounds, the outcompetition of pathogens, the strengthening of the intestinal barrier, and the stimulation and regulation of the immune system (see recent reviews 1, 2, 3, 4, 5, 6, 7). Most of these functions are interconnected and tightly intertwined with human physiology. For example, products of microbial fermentation, such as short-chain fatty acids, represent essential substrates for intestinal cells and play important roles in immunomodulatory processes, such as T cell differentiation, which, in turn, may affect the gut microbiome.
- Finally, in the reviewer’s opinion, for better tracking and understanding of the manuscript, Figure 2 should be before than Table 1.
Thank you, we have changed the figures order and added other figures as requested by other Reviewers (see below).
Reviewer 2 Report
The authors present an interesting review on the interaction of the intestinal microbiota and glucosinolates. The review includes generalities about glucosinolates and their derivatives, bacteria with myrosinase activity and the relationship between the microbiota and glucosinolates.
Section 3 seems too short to be a separate section, it is suggested that it could be added to the introduction.
Section 4 is too general and its content is not related to the main topic of the article. On the other hand, it is not understood why there is a subsection a. In any case, this subsection talks mainly about prebiotics and very especially about fiber, which is not related to the subject of the article either. It is suggested the deletion of section 4 and its subsection a.
English language needs an important revision.
Author Response
We are pleased to submit for evaluation our review titled “The metabolism of glucosinolates by gut microbiota.” (by Kalina Sikorska-Zimny and Luciano Beneduce), which has gone under the review in Nutrients, has been rejected from publication (December 2020), than editors encourage authors for publication (May 2021).
Note: "This manuscript is a resubmission of "nutrients-1027223" for special issue "Gut Microbiota in Human Health and Diseases".
Below we had answered all remarks made by the Reviewers.
In the body of manuscript according to editor’s demanding we have left all previous marked changes and made a revisions to the manuscript, marked up using the “Track Changes” function.
K.Sikorska-Zimny
L. Beneduce
Reviewers (see below).
The authors present an interesting review on the interaction of the intestinal microbiota and glucosinolates. The review includes generalities about glucosinolates and their derivatives, bacteria with myrosinase activity and the relationship between the microbiota and glucosinolates.
Section 3 seems too short to be a separate section, it is suggested that it could be added to the introduction.
Thank you, we have moved the section 3 to the second as a short explanation of gls pathways in human digestion track.
Section 4 is too general and its content is not related to the main topic of the article. On the other hand, it is not understood why there is a subsection a. In any case, this subsection talks mainly about prebiotics and very especially about fiber, which is not related to the subject of the article either. It is suggested the deletion of section 4 and its subsection a.
English language needs an important revision.
Thank you for your comment, we consider it appropriate to present this information as a justification for putting more emphasis on the effect of vegetable consumption and as an introduction to the chapter dealing directly with the effects of GLS on HGM.
Reviewer 3 Report
My comments are in the attached file

Author Response
We are pleased to submit for evaluation our review titled “The metabolism of glucosinolates by gut microbiota.” (by Kalina Sikorska-Zimny and Luciano Beneduce), which has gone under the review in Nutrients, has been rejected from publication (December 2020), than editors encourage authors for publication (May 2021).
Note: "This manuscript is a resubmission of "nutrients-1027223" for special issue "Gut Microbiota in Human Health and Diseases".
Below we had answered all remarks made by the Reviewers.
In the body of manuscript according to editor’s demanding we have left all previous marked changes and made a revisions to the manuscript, marked up using the “Track Changes” function.
K. Sikorska-Zimny
L. Beneduce
REWIEVER 3
Comments to paper The metabolism of glucosinolates by gut microbiota
Interesting paper but chemical structures are missing, and they would help to understand what is really being talked about.
This is a review paper, this should be indicated somewhere.
- Figure 1 is not OK, it should be sharper, SO3- is not connected to the rest of the molecule, and it cannot SO3-. There are papers with nice schemes of this transformation, showing chemically what happens.
Thank you we have corrected the figure 1, by improving the scheme and merged it with the previous figure 2, to have a comprehensive scheme.
- The enzyme should have a number, E.C. classification.
Thank you for your comment, although in line 48 (the reviewed version) we have written:” The enzymatic reaction between the myrosinase enzymes (EC 3.2.1.147, thioglucoside glucohydrolase, sinigrinase, sinigrase, MYR) and GLS leads to the production of GLS derivatives.”
We have also add th enzyme no to the figure 1.
- Line 86 refers chemical structure of glucosinolates without showing the chemical structure, it would be easier to have a figure showing what this is chemically. During the paper several different chemical structures are referred but there are no images to facilitate the reading an understanding what happens to the structures during the enzymatic hydrolysis.
Thank you for your comment, we have added the structures in the figure (fig 1).
- Line 90: aliphatic, aromatic and indole groups where are they in the glucosinolate structure? It would be good to see where is the «R» group that can have all these hypotheses.
Thank you for your comment, we have modified the figure 1 and showed the molecular structures of different GLS derivatives.
- Line 95-100: confusing without seeing structures. Would it be possible to have a figure?
Thank you for your comment, we have added a new figure (Fig 2).
- Line 120: Can´t figure 2 also have chemical structures?
Thank you for your comment, we have corrected the figure.
- Line 137 and 144 classification c) and d) are the same? Thiocyanates, both?
Thank you for your observation: we have added the missing “iso-“ where needed.
- Line 240: In the human feeding studies it is always difficult to have straight conclusions, when the referred papers in their studies gave cruciferous in the human diet, probably this was mixed with other foods, how to separate the effects?
We agree with the Reviewer comment. The paragraph indeed had the purpose to stress the point and only showed the conclusion about the general effect on microbial diversity of HGM related to cruciferous rich diet. We added a sentence in the paragraph to further evidence the difficulties to have straight conclusions. As we reported in the manuscript, we think that further studies, both in vitro and in vivo, so as non-cultivation along with cultivation microbial methods, must be combined to address to more robust conclusions.
- Line 263 and others: Latin words should be in italic, «in vivo» etc.
Thank you, we have corrected the Latin words.
- Myrosinases always catalyze the hydrolysis in the chemical bond of the glucose moiety, and produce isothiocyanates, If so what has glucoraphanin in particular to be mentioned? It is missing the chemical differences between all these compounds. Probably the enzyme has an active site that requires some chemical confirmation for the substrates. Is it already known?
We are not sure to which part of the manuscript the Reviewer was referring to: if it is the line 287, we did a minor correction (glucoraphanin instead of sulphoraphane) that was by mistake reported. In general Glucoraphanin is the most studied GLS, along with sinigrin, for many reasons (average content in Brassicas, proven pro-healthy functions of their derivative etc.) Therefore, it is often used as a “model” GLS In many in vitro and in vivo studies. About microbial myrosinases activitiy, as reported in the manuscript, different GLS can be metabolized to different products.
- Line 308: «the genetic and biochemical basis for activation of glucosinolates to isothiocyanates», activation? What does this mean Where does it occur?
Thank you for the comment: the literature article cited in the context of the sentence used the term “activation” due to the pro-healthy derivate isothiocyanates that are considered dietary beneficial. They clarified the genetic mechanisms that lead to the activation of GLS conversion to isothiocyanates in Bacteriodes Thetaiotaomicron, through the activation of key operone and expression of the relative genes.
- Figure 3 need to be improved.
We improved the figure according to the Reviewer comment.
Reviewer 4 Report
This review deals with the metabolism of glucosinolates by gut microbiota. An important impact on human health has been reported by GLS derivatives in the presence of myrosinase enzyme, since they present anti-inflammation and anti-cancer properties. The topic of the reviewed manuscript is suitable for the journal, but the English should be improved in all the manuscript.
Several points should be addressed before being published in Nutrients:
- Please, improve the sentence (lines 58-62). Please, avoid using such long sentences. Please review the entire manuscript
- Please, improve the quality of the figures
- Please, improve the conclusions section
- The list of references must be updated. Include all done in this field in the last 3 years. Making reference to recent work in the field is particularly key to highlight the current context of the present manuscript and to make it more comprehensive and to highlight the novelty to the readers as well as its assessment and contribution to the field. Please, address this request by adding new critical analysis and not by simply citing papers published in this subject.
Author Response
We are pleased to submit for evaluation our review titled “The metabolism of glucosinolates by gut microbiota.” (by Kalina Sikorska-Zimny and Luciano Beneduce), which has gone under the review in Nutrients, has been rejected from publication (December 2020), than editors encourage authors for publication (May 2021).
Note: "This manuscript is a resubmission of "nutrients-1027223" for special issue "Gut Microbiota in Human Health and Diseases".
Below we had answered all remarks made by the Reviewers.
In the body of manuscript according to editor’s demanding we have left all previous marked changes and made a revisions to the manuscript, marked up using the “Track Changes” function.
K. Sikorska-Zimny
L. Beneduce
REWIEVER 4
This review deals with the metabolism of glucosinolates by gut microbiota. An important impact on human health has been reported by GLS derivatives in the presence of myrosinase enzyme, since they present anti-inflammation and anti-cancer properties. The topic of the reviewed manuscript is suitable for the journal, but the English should be improved in all the manuscript.
Thank you for your comment we have corrected the English language along the manuscript.
Several points should be addressed before being published in Nutrients:
- Please, improve the sentence (lines 58-62). Please, avoid using such long sentences. Please review the entire manuscript
Thank you we have shorten the sentences and corrected in whole manuscript.
- Please, improve the quality of the figures
We have modified all figures and added new ones as requested also by other Reviewers.
- Please, improve the conclusions section
Thank you we have improved conclusion chapter, by adding new comments also related to more recent publications (see below).
- The list of references must be updated. Include all done in this field in the last 3 years. Making reference to recent work in the field is particularly key to highlight the current context of the present manuscript and to make it more comprehensive and to highlight the novelty to the readers as well as its assessment and contribution to the field. Please, address this request by adding new critical analysis and not by simply citing papers published in this subject.
Thank you , we have updated the references, adding 6 new articles form the last 3 years, some of them in press (e.g. reference n. 45 was accepted in the last month). The manuscript text has been updated accordingly with the new references cited.
Round 2
Reviewer 2 Report
The paper has been sufficiently improved to warrant publication in Nutrients
This manuscript is a resubmission of an earlier submission. The following is a list of the peer review reports and author responses from that submission.
Round 1
Reviewer 1 Report
The manuscript is easy to follow and it is relatively well organized. However, the concern I have with the manuscript is the lack of novelty and the depth of the review.
Table 1 takes one printed page, but I find that it adds little value to the review. The title is also confusing. Do authors mean these are main glucosinolates in Brassicas or in general? Instead, I would prefer to see a table summarizing metabolism of GSL by microbiota.
To conclude, I do believe the subject of gut metabolism of GSL is very important and does lack knowledge. I think authors did a great job laying a foundation for good quality paper and can significantly improve the manuscript by adding in depth analysis of existing data. For example, authors may consider exploring an indirect effect of GSL on gut microbes, non-enzymatic hydrolysis of GSL, or non-hydrolytic reactions in gut environment such as formation of complexes with proteins.
Author Response
Corresponding author
Kalina Sikorska-Zimny
kalinasikorskazimny@gmail.com
Dear Editor-in-Chief of Nutrients,
We are pleased to submit for evaluation our review titled “The metabolism of Glucosinolates by gut microbiota.” (by Kalina Sikorska-Zimny and Luciano Beneduce), which has undergone the first review in Nutrients.
Below we had answered all remarks made by the Reviewer 1.:
The manuscript is easy to follow and it is relatively well organized. However, the concern I have with the manuscript is the lack of novelty and the depth of the review.
Thank you for your comments. We had deepened our studies (including new and improved tables and graphs) and highlighted the novelty in a context of GLS and human gut.
Table 1 takes one printed page, but I find that it adds little value to the review. The title is also confusing. Do authors mean these are main glucosinolates in Brassicas or in general? Instead, I would prefer to see a table summarizing metabolism of GSL by microbiota.
We had rechanged the table by adding a column dedicated to possible formulated from GLS – compound.
To conclude, I do believe the subject of gut metabolism of GSL is very important and does lack knowledge. I think authors did a great job laying a foundation for good quality paper and can significantly improve the manuscript by adding in depth analysis of existing data. For example, authors may consider exploring an indirect effect of GSL on gut microbes, non-enzymatic hydrolysis of GSL, or non-hydrolytic reactions in gut environment such as formation of complexes with proteins.
We had placed the table 2 where possible relation between microbial and GLS is presented.
It has been rearranged.
Yours faithfully,
K. Sikorska-Zimny
L. Beneduce
TO WHOM IT MIGHT CONCERN
I hereby state that the manuscript entitled “The metabolism of Glucosinolates by gut microbiota.” by Kalina Sikorska-Zimny (et al.) was professionally proofread before submission to improve the English.
Your sincerely,
Colin Walker
TEACH U
Kwiatowa 10/1
Warszawa, 02-579
NIP: 7122785425
REGON: 060204026
https://www.e-korepetycje.net/englishconversation/jezyk-angielski

Reviewer 2 Report
Overview/summary of the manuscript
The manuscript “The metabolism of Glucosinolates by gut microbiota.” by Sikorska-Zimny K et al. is a review focused on the recent advances about the role of microbiota on the glucosinolates metabolism. The authors first make an introduction about what are and where are found the glucosinolates. Then they present the glucosinolates structure and their derivatives. Next, the authors talk about the human microbiota and the influence of some plants components on the microbiota composition. Finally, the authors show the evidences about the influence of glucosinolates and their derivatives on human gut microbiota, and how some of bacterial genera of microbiota, that have myrosinase activity, could contribute to the hydrolysis of glucosinolates and formation of their active derivatives. The authors conclude that the microbiota could have a significant impact increasing the bioavailability of glucosinaltes derivatives. A potential future of metabolic engineering isolating the most promising bacterial strain and use it as a probiotic is proposed.
Broad comments
In the opinion of this reviewer, the review topic is interesting and topical, but the manuscript is not well organized, it is difficult to follow, and in addition there are errors in concepts in some parts, for example sections 3 and 4. In addition, the main object of review constitutes a small part of the manuscript, it should be expanded this part of manuscript. The manuscript could improve with some schemes (e.g. classifiaction of glucosinolates derivatives). Further, the authors are not careful about the abbreviations, they do not always introduce abbreviations when first citing (e.g. line 143 SCF-acids and then in line 170 appears short-chain fatty acids). Finally, the article would benefit from a close editing. It is possible that the difficulty to follow the author’s argument could due to the many stylistic and grammatical errors.
Specific comments
- line 13 abstract and line 28 in introduction: “most abundant in brassicas” and “mostly present in plants of family Brassicaceae” Brassicae are the only vegetables that there are glucosinolates, this make them different.
- line 40: the references 5, 6 seem wrong, they refer us to those biological effects.
- Table 1 is not mentioned in the text
- A scheme for the glucosinolates derivatives it would be more understanding.
- line 24: indoles are not stables isothiocyanates
-Section 3: “the role of intestine digestion ……..Gut” and “Fermentation food (digestion…), apart from that the last sentence is incomprehensible, there are a wrong of concept. Digestion take place in the small intestine by host digestive enzymes (proteases, lipases, amylase disaccharidases…), the vegetable material that cannot be digested in the small intestine by host digestive enzyme will arrive to the large intestine (mostly fiber but other compounds, as e.g. GLS) where they can be fermented by microbiota. Therefore, the digestion does not take place by microbiota!!!
- Section 4: this section is difficult to understanding, because e.g. you talks about short-chain fatty acids without any contextualization. In sub-section a) Fermentable carbohydrates, the authors talk about starch, when the starch is the most digestible carbohydrate, only a small fraction of starch are non-digestible and is fermented by microbiota. Here the pro-bifidogenic effect that the authors refers is due to the fiber, mainly soluble fiber. Then sub-section b) prebiotics, the compounds that authors cite here are fiber!!! Fiber has a proven probiotic effect. The sub-sections a and b could be resumed in one referring to a "non-digestible carbohydrates" (that is, fiber).
-Sections 3 and 4 should be almost entirely rewritten
Author Response
Corresponding author
Kalina Sikorska-Zimny
kalinasikorskazimny@gmail.com
Dear Editor-in-Chief of Nutrients,
We are pleased to submit for evaluation our review titled “The metabolism of Glucosinolates by gut microbiota.” (by Kalina Sikorska-Zimny and Luciano Beneduce), which has undergone the first review in Nutrients.
Below we had answered all remarks made by the Reviewer 2.
Overview/summary of the manuscript
The manuscript “The metabolism of Glucosinolates by gut microbiota.” by Sikorska-Zimny K et al. is a review focused on the recent advances about the role of microbiota on the glucosinolates metabolism. The authors first make an introduction about what are and where are found the glucosinolates. Then they present the glucosinolates structure and their derivatives. Next, the authors talk about the human microbiota and the influence of some plants components on the microbiota composition. Finally, the authors show the evidences about the influence of glucosinolates and their derivatives on human gut microbiota, and how some of bacterial genera of microbiota, that have myrosinase activity, could contribute to the hydrolysis of glucosinolates and formation of their active derivatives. The authors conclude that the microbiota could have a significant impact increasing the bioavailability of glucosinaltes derivatives. A potential future of metabolic engineering isolating the most promising bacterial strain and use it as a probiotic is proposed.
Broad comments
In the opinion of this reviewer, the review topic is interesting and topical, but the manuscript is not well organized, it is difficult to follow, and in addition there are errors in concepts in some parts, for example sections 3 and 4.
Thank you for your comment. We had reorganized both sections with introduction of basics of human digestive tract incl scheme.
In addition, the main object of review constitutes a small part of the manuscript, it should be expanded this part of manuscript.
We had expended this part of manuscript and add table (no.2).
The manuscript could improve with some schemes (e.g. classifiaction of glucosinolates derivatives).
We have provided such a scheme2 (no. 2 and 3).
Further, the authors are not careful about the abbreviations, they do not always introduce abbreviations when first citing (e.g. line 143 SCF-acids and then in line 170 appears short-chain fatty acids).
Thank you for pointing a mistake. We had gone throughout all the manuscript and corrected mistakes and missing abbreviations.
Finally, the article would benefit from a close editing. It is possible that the difficulty to follow the author’s argument could due to the many stylistic and grammatical errors.
The manuscript had been also reviewed by English native speaker (vide confirmation letter).
Specific comments
- line 13 abstract and line 28 in introduction: “most abundant in brassicas” and “mostly present in plants of family Brassicaceae” Brassicae are the only vegetables that there are glucosinolates, this make them different.
Thank you for your tip. We had corrected the sentence to a family of Brassicaceae
- line 40: the references 5, 6 seem wrong, they refer us to those biological effects.
Reference 5 -Lafarga et al. are referring to the cancer (possible factors influencing) and reference 6 -García-Saldaña et al. pointed on osteoarthritis (that might be delay with GLS).
- Table 1 is not mentioned in the text
Thank you - we have place it in a text.
- A scheme for the glucosinolates derivatives it would be more understanding.
Thank you - we have added a scheme (as fig 2).
- line 24: indoles are not stables isothiocyanates
Thank you for pointing! We have placed the correct sentence and rearranged a subchapter
-Section 3: “the role of intestine digestion ……..Gut” and “Fermentation food (digestion…), apart from that the last sentence is incomprehensible, there are a wrong of concept. Digestion take place in the small intestine by host digestive enzymes (proteases, lipases, amylase disaccharidases…), the vegetable material that cannot be digested in the small intestine by host digestive enzyme will arrive to the large intestine (mostly fiber but other compounds, as e.g. GLS) where they can be fermented by microbiota. Therefore, the digestion does not take place by microbiota!!!
Thank you for your comment. Due to the topic of an article as well as focusing on a main topic and possible allegation of deviating from the topic we had omitted all digestion process. In improved manuscript we included schemes with correct digestion conditions.
- Section 4: this section is difficult to understanding, because e.g. you talks about short-chain fatty acids without any contextualization.
Thank you for your comment. We had explained the role of short-chain fatty acids in the text.
In sub-section a) Fermentable carbohydrates, the authors talk about starch, when the starch is the most digestible carbohydrate, only a small fraction of starch are non-digestible and is fermented by microbiota. Here the pro-bifidogenic effect that the authors refers is due to the fiber, mainly soluble fiber. Then sub-section b) prebiotics, the compounds that authors cite here are fiber!!! Fiber has a proven probiotic effect. The sub-sections a and b could be resumed in one referring to a "non-digestible carbohydrates" (that is, fiber).
We had corrected and rearranged the fibre sub-chapter to more clear and correct structure.
-Sections 3 and 4 should be almost entirely rewritten
It has been rearranged.
Yours faithfully,
K. Sikorska-Zimny
L. Beneduce
TO WHOM IT MIGHT CONCERN
I hereby state that the manuscript entitled “The metabolism of Glucosinolates by gut microbiota.” by Kalina Sikorska-Zimny (et al.) was professionally proofread before submission to improve the English.
Your sincerely,
Colin Walker
TEACH U
Kwiatowa 10/1
Warszawa, 02-579
NIP: 7122785425
REGON: 060204026
https://www.e-korepetycje.net/englishconversation/jezyk-angielski

Round 2
Reviewer 1 Report
The manuscript was definitely improved with the addition of new tables and figure. There are few minor editorial issues in the manuscript. For example, Table 2 is cited before Table 1.
Author Response
Reviewer: The manuscript was definitely improved with the addition of new tables and figure. There are few minor editorial issues in the manuscript. For example, Table 2 is cited before Table 1.
Answer: Thank you very much, we had run throughout all manuscript improving writing mistakes and add the citation of table 1 as first before table 2.
Reviewer 2 Report
The authors have made an effort to improve the manuscript. They have modified Table 1 and introduced different figures and a new table. Despite this, much of the comments this reviewer made have not been changed or improved. Consequently, I still think that in general the manuscript continues to have serious defects. Below I set out those aspects that have not improved:
- Further, the authors are not careful about the abbreviation. Abbreviations without their full name still appear: eg line 79 AITC or that of line 178 SCF-acids mentioned in the previous review!!
- Has not changed: line 13: “most abundant in brassicas” and line 29 “mostly present in plants of family Brassicaceae” Could the authors tell me what other vegetables frequently consumed in human nutrition contain glucosinolates?
- The authors are wrong when they explain references 5 and 6 of line 40: Lafarga, T.et al. Effects of thermal and non-thermal processing of cruciferous vegetables on glucosinolates and its derived forms. J Food Sci Technol 2018, 55(6), 1973–1981. And García-Saldaña,J.S et al. Separationand purification of sulforaphane (1-isothiocyanato-4-(methylsulfinyl) butane) from broccoli seeds by consecutive steps of adsorption-desorption-bleaching. J Food Eng 2018, 273, 162-170. please read them, they are methodological !!!!
- The conceptual errors in Chapter 3 follow, the sentence: “The process of food digestion is strictly dependent on the quantitative and qualitative composition of the microorganisms that inhabit the gut” it is totally false!!!! Please, if the authors do not know physiology or nutrition perhaps it would be better for them to omit this section before saying barbarities. Digestion does not depend on microorganisms. If it was like that, why do we have pancreatic digestive enzymes, such as amylase, lipase and proteases, and even more the enzymes secreted by the intestinal mucosa, e.g. lactase, glucoamylase, proteases….? The scheme that the authors have introduced is superfluous and does not contribute in anything or solve these conceptual problems, I think it needs to be eliminated. As the authors comment on line 234 referring to the concept of prebiotics (it should also be noted that this would not only be the definition of prebiotics it would be the global definition of fiber) here we have the implicit concept of digestion ("non-digestible"), which is different from that of hydrolysis or fermentation of components by the microbita at the level of the large intestine. In addition, the scheme also provides nothing with respect to glucosinolates (the glucosinolates are not digested). Glucosinolates will be hydrolyzed, in part, due to the release of myrosinase because chewing process. The glucosinolates derivatives will suffer few changes in is passage through the stomach (except I3C which tends to form polymers at acidic pH) and small intestine. In this way, in the small intestine will absorbed those glucosinoltes derivatives previous formed by the action of myrosinase. Finally, those glucosinolates that arrive intact in the large intestine are those that can be hydrolyzed by the myrosinase of the microbiota. So, this scheme does not illustrate this cleary!!
Other comments:
- Line155: The title of section “d) Thiocyanates” are missing
- line 236: the compounds cited are food natural constituent included in fiber concept and classification. Other confuse concept for the authors!!!; Change "galacto-oligosaccharides" to "galacto-oligosaccharides-GAL" for coherence with "fructo-ogigosaccharises-FOS" or the inverse.
- Figure 4: Legend must be changed by one that best describe it, as eg: "Scheme of GLS breakdown products by human microbiota and their adsorption in large intestine....."
- Table 2: Shows a series of abbreviations referring to glucosinolates and their derivatives which are not made explicit anywhere and which is very difficult to understand. On the other hand, in Table 1 there are the full names without the abbreviation. In one or another place, the whole name should be related to the abbreviation.
Author Response
Reviewer: The authors have made an effort to improve the manuscript. They have modified Table 1 and introduced different figures and a new table. Despite this, much of the comments this reviewer made have not been changed or improved. Consequently, I still think that in general the manuscript continues to have serious defects. Below I set out those aspects that have not improved:
We went through all the manuscript, following reviewer comments and changed all parts that still needed to be improved. Many parts were amended, and major changes were done in the more critic parts. We tracked all these amendments, and we answered here below to the major and minor comments.
- Further, the authors are not careful about the abbreviation. Abbreviations without their full name still appear: eg line 79 AITC or that of line 178 SCF-acids mentioned in the previous review!!
Thank you, we had explained all abbreviation used in text carefully.
- Has not changed: line 13: “most abundant in brassicas” and line 29 “mostly present in plants of family Brassicaceae” Could the authors tell me what other vegetables frequently consumed in human nutrition contain glucosinolates?
Brassicaceae are the family that includes genus Brassica but also other genus that has GLS - Sinapsis L. Therefore, we used Brassicaceae that points on mustards, moutardes, crucifers but then highlighted that we had focused on Brassica L.
We changed the sentence in order to better clarify it in the text.
- The authors are wrong when they explain references 5 and 6 of line 40: Lafarga, T.et al. Effects of thermal and non-thermal processing of cruciferous vegetables on glucosinolates and its derived forms. J Food Sci Technol 2018, 55(6), 1973–1981. And García-Saldaña,J.S et al. Separation and purification of sulforaphane (1-isothiocyanato-4-(methylsulfinyl) butane) from broccoli seeds by consecutive steps of adsorption-desorption-bleaching. J Food Eng 2018, 273, 162-170. please read them, they are methodological !!!!
Thank you for your comment, we had referred to basic information given/cited by the other authors and do not going deeply into methodology of origin examination. Now we have changed the references according with reviewer’s comment. We referred to the author that primarily stated the subject of the citation.
- The conceptual errors in Chapter 3 follow, the sentence: “The process of food digestion is strictly dependent on the quantitative and qualitative composition of the microorganisms that inhabit the gut” it is totally false!!!! Please, if the authors do not know physiology or nutrition perhaps it would be better for them to omit this section before saying barbarities. Digestion does not depend on microorganisms. If it was like that, why do we have pancreatic digestive enzymes, such as amylase, lipase and proteases, and even more the enzymes secreted by the intestinal mucosa, e.g. lactase, glucoamylase, proteases….?
The conceptual errors of chapter 4 (not 3) have been removed and authors amended the first part, where they better explained the basic concept of the actual role of gut microbiota in digestion processes, and other functions in human health. We also added new references when needed.
-The scheme that the authors have introduced is superfluous and does not contribute in anything or solve these conceptual problems, I think it needs to be eliminated.
We deleted the scheme and decided only to rewrite the text section, so the figure scheme has been removed.
-As the authors comment on line 234 referring to the concept of prebiotics (it should also be noted that this would not only be the definition of prebiotics it would be the global definition of fiber) here we have the implicit concept of digestion ("non-digestible"), which is different from that of hydrolysis or fermentation of components by the microbita at the level of the large intestine.
We have included the section of prebiotics in the non-digestible carbohydrates with addition of examples of NDCs and basic explanation of its correlation with human gut microbiota.
-In addition, the scheme also provides nothing with respect to glucosinolates (the glucosinolates are not digested).
We delete a scheme and decided only to rewrite the text section, so the figure scheme has been removed.
-Glucosinolates will be hydrolyzed, in part, due to the release of myrosinase because chewing process. The glucosinolates derivatives will suffer few changes in is passage through the stomach (except I3C which tends to form polymers at acidic pH) and small intestine. In this way, in the small intestine will absorbed those glucosinoltes derivatives previous formed by the action of myrosinase. Finally, those glucosinolates that arrive intact in the large intestine are those that can be hydrolyzed by the myrosinase of the microbiota. So, this scheme does not illustrate this cleary!!
We delete the scheme 3 and fulfilled the chapter 3 with basic information of how GLS’s conversion processes are occurring in human digestive tract.
Other comments:
- Line155: The title of section “d) Thiocyanates” are missing
Thank you we have added the sub-par, by splitting thiocyanates and isothiocyanates in 2 different sections to be coherent with the figure 2 as well.
- line 236: the compounds cited are food natural constituent included in fiber concept and classification. Other confuse concept for the authors!!!; Change "galacto-oligosaccharides" to "galacto-oligosaccharides-GAL" for coherence with "fructo-ogigosaccharises-FOS" or the inverse.
We have corrected galacto-oligosaccharides into galacto-oligosaccharides-GAL and fulfill the sentence.
- Figure 4: Legend must be changed by one that best describe it, as eg: "Scheme of GLS breakdown products by human microbiota and their adsorption in large intestine....."
We have improved the tile of figure 4 according to the suggestion.
- Table 2: Shows a series of abbreviations referring to glucosinolates and their derivatives which are not made explicit anywhere and which is very difficult to understand. On the other hand, in Table 1 there are the full names without the abbreviation. In one or another place, the whole name should be related to the abbreviation.
We have reported the abbreviation of every GLS and derivative cited in the paper in the table 1 . We corrected Table 2 using coherent abbreviations, and consequently changed the caption and undertext to improve the clarity of the information given. Also, some minor amendments were made to the table 2 (genus and species putted correctly in the column heading, and strains reported between brackets).